# SIKeD: Self-guided Iterative Knowledge Distillation for mathematical reasoning

## Abstract

Large Language Models (LLMs) can transfer their reasoning skills to smaller models by teaching them to generate the intermediate reasoning process required to solve multistep reasoning tasks. While LLMs can accurately solve reasoning tasks through a variety of strategies, even without fine-tuning, smaller models are not expressive enough to fit the LLMs distribution on all strategies when distilled and tend to prioritize one strategy over the others. This reliance on one strategy poses a challenge for smaller models when attempting to solve reasoning tasks that may be difficult with their preferred strategy. To address this, we propose a distillation method SIKeD: **S**elf-guided **I**terative **K**nowledge **D**istillation[1], where the LLM teaches the smaller model to approach a task using different strategies and the smaller model uses its self-generated on-policy outputs to choose the most suitable strategy for the given task. The training continues in a *self-guided* iterative manner, where for each training iteration, a decision is made on how to combine the LLM data with the self-generated outputs. Unlike traditional distillation methods, SIKeD allows the smaller model to learn *which* strategy is suitable for a given task while continuously learning to solve a task using different strategies. Our experiments on various mathematical reasoning datasets show that SIKeD significantly outperforms traditional distillation techniques across smaller models of different sizes.

## 1 Introduction

Large language models (LLMs), with tens to hundreds of billions of parameters, generally outperform smaller models (with billions of parameters or fewer) in a variety of reasoning tasks Touvron et al. (2023); Achiam et al. (2023). One notable strength of large models is their ability to reason and perform multistep reasoning tasks, often considered an important aspect of intelligence Gómez-Veiga et al. (2018). However, the significant size and computational demands of these large models present several challenges. For example, LLaMA3 models Touvron et al. (2023) are trained using clusters of 24,000 GPUs, limiting their accessibility to many researchers and practitioners.

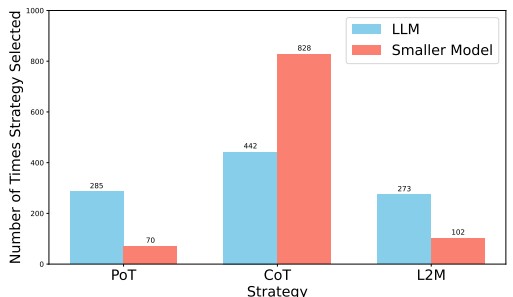

Figure 1: **Histogram of strategy choices for the LLM and the smaller model.** LLM tends to select several reasoning strategies, but the smaller model is biased towards one strategy. The comparison was done on 1000 data points randomly sampled from the GSM8K train set.

To bridge this gap, a key approach involves teaching smaller models to replicate the knowledge of a larger model, often referred to as *knowledge distillation* Hinton (2015). Typically, smaller models can be taught to replicate the multistep reasoning capabilities of larger models by incorporating a set of intermediate sequences (Kim & Rush, 2016; Shridhar et al., 2023). However, these intermediate steps can be derived from a number of different strategies, such as Chain of Thought (CoT) Wei et al. (2022), Subquestion Decomposition Shridhar et al. (2022); Zhou et al. (2023), and Program of Thoughts (PoT) Chen et al. (2023), among others. A

---

[1]SIKeD is pronounced as "psyched"

viable solution is to distill these reasoning capabilities into smaller models either by distilling individual strategies (Magister et al., 2023; Shridhar et al., 2023; Hsieh et al., 2023) or by incorporating multiple strategies simultaneously (Chenglin et al., 2023; Zhu et al., 2024). Although smaller models have demonstrated impressive performance when distilled with a single strategy, they often struggle to master multiple strategies equally well. An example is presented in Figure 1 where a larger model can use multiple strategies to generate data but upon distilling, a smaller model tends to favor one over the others. This is because reasoning through a variety of strategies tends to emerge as a result of scaling language models, making it difficult for smaller models to replicate this behavior Lyu et al. (2024).

On the other hand, learning to solve a task using multiple strategies can help smaller models overcome the limitations of relying on a single approach. However, a key challenge arises when, despite being trained on a fixed dataset containing various strategies, a distribution mismatch occurs between the data generated by the LLM and the outputs produced by the smaller model during inference. This mismatch can hinder the ability of the smaller model to generalize across different reasoning strategies. This issue, often discussed in imitation learning (Pomerleau, 1991; Ross & Bagnell, 2010), results in the student model consistently choosing one strategy, even when a different approach would be more appropriate. As a result, the student generates outputs with strategy choices that are highly unlikely to match those produced by the teacher.

To address this challenge, we introduce our distillation methodology, `SIKeD`: **S**elf-guided **I**terative **K**nowledge **D**istillation. The process begins with the LLM teaching the smaller model to approach tasks using a variety of reasoning strategies, providing a strong foundation for the smaller model to understand different problem-solving approaches. However, due to inherent biases and its limited capacity, the smaller model may still struggle to match the LLM's distribution of strategy choices effectively. To resolve this, we take inspiration from constructivist learning theory Narayan et al. (2013), where the learner builds knowledge during the "assimilation phase" and refines their understanding during the "accommodation phase" to incorporate new insights. We propose generating outputs using the smaller model in an on-policy setup and selecting the best strategies for the task. By mixing the LLM-generated data with self-generated outputs, we leverage the strengths of both datasets. We iteratively fine-tune the smaller model allowing it to recognize strategies that it learned from the LLM but did not initially apply. With this approach, we align the smaller model with its own learned knowledge rather than forcing its distribution to mirror that of the LLM's.

Our proposed method extends beyond traditional one-step distillation, as each iteration of `SIKeD` leads to an updated policy that better grasps new information. We repeat multiple iterations of `SIKeD` based on the accuracy-cost tradeoff (does the improvement justify the cost of another iteration), allowing for continuous refinement and improvement of the model's reasoning capabilities. We demonstrate the effectiveness of `SIKeD` on several mathematical reasoning tasks using models with fewer than 7 billion parameters.[2] On four mathematical datasets—GSM8K Cobbe et al. (2021), SVAMP Patel et al. (2021), ASDiv Miao et al. (2020), and MultiArith Roy & Roth (2016)—our approach achieves improvements of up to +5 points over traditional distillation strategies. Additionally, we show that multiple rounds of `SIKeD` allow the model to select the appropriate strategy for a given problem, while traditional distillation using LLM's data tends to leave it biased.

## 2 PRELIMINARIES: LLM BASED DISTILLATION

**Problem Setup**   We consider the standard setup of LLM-based distillation (also referred to as *knowledge distillation*), where the data is sampled from the larger model (LLM) with intermediate reasoning and the smaller model is fine-tuned (distilled) over it Shridhar et al. (2023); Magister et al. (2023). Two auto-regressive sequence models are involved in the process: a larger model or the LLM denoted as $p_L$ and a smaller model to be distilled as $p_{sm}^{\theta}$ (with learnable parameters $\theta$). In this work, we consider a reasoning dataset $\mathcal{D}$ consisting of a question $q_i$ and a numerical answer $a_i$ pairs for $n$ data points, i.e. $i \in \{1, \ldots, n\}$. Since our work focuses on improving the reasoning in smaller models by teaching them to solve a variety of reasoning strategies, we consider three reasoning strategies in this work: Chain-of-Thought (`CoT`), Least-to-Most (`L2M`), and Program-of-Thought (`PoT`). For a specific *reasoning strategy*, denoted as $s \in S$, we generate the reasoning chain or *rationale*, denoted

---

[2]We acknowledge that "smaller model" is a relative term, and we consider models with fewer than 7 billion parameters to be smaller models.

as $r_i$ leading to the final answer as: $r_i \sim p_L(. \mid \mathtt{pr}_s, q_i)$, where, $\mathtt{pr}_s$ represents the strategy-specific prompt, and $s \in \{\mathtt{CoT}, \mathtt{L2M}, \mathtt{PoT}\}$. Prompts used are provided in Appendix A.

## 2.1 LLM BASED DISTILLATION

We add the generated rationales to the the dataset $\mathcal{D}$ to create an initial training dataset $\mathcal{D}_{\text{LLM}}$ consisting of a quadruple of $\{q_i, a_i, s, r_i\}$ for each data point. We perform a data filtering by extracting the final answer $\hat{a}_i$ from the generated rationale $r_i$ and comparing it with the ground truth answer $a_i$. We discard all samples that do not match, i.e., we keep samples where $\hat{a}_i = a_i$. This filtering process eliminates incorrect rationales, ensuring that only high-quality data is used for distilling the smaller models.

We start the distillation process by training the smaller model with the created dataset $\mathcal{D}_{\text{LLM}}$. The question $q_i$ is provided as input, and the smaller model $p_{\text{sm}}^\theta$ (with learnable parameters $\theta$) is first instructed to generate the strategy $s$, followed by the rationale $r_i$ that leads to the final answer $a_i$. The loss $\mathcal{L}_{\text{L}}(\theta)$ is defined as:

$$\mathcal{L}_{\text{L}}(\theta) = -\mathbb{E}_{(q_i, s, r_i) \sim \mathcal{D}_{\text{LLM}}} \left[ \log p_{\text{sm}}^\theta (s \mid q_i, I) + \sum_{t=1}^{M} \log p_{\text{sm}}^\theta (r_{i,t} \mid r_{i,<t}, s, q_i, I) \right], \quad (1)$$

where $M$ represents the number of tokens decoded over time $t$ in an autoregressive manner, and $I$ is the instruction used during fine-tuning. Note that this is analogous to traditional knowledge distillation from LLMs except that we make a strategy choice before generating rationales.

**Limitations** Training solely on LLM-generated data $\mathcal{D}_{\text{LLM}}$ can lead to a distribution mismatch between the training data and the smaller model's own output distribution. Specifically, the larger model due to its larger capacity, may produce correct reasoning across multiple strategies that the smaller model can find difficult to replicate directly Agarwal et al. (2024). A comparison of the strategy selected by the LLM and the smaller model on 1K samples is presented in Figure 1. The smaller model performs poorly when generating outputs on its own, as the training data distribution $P_{\text{train}}(x)$ is different from the model's output distribution $P_{\text{sm}}^\theta(x)$ as $P_{\text{train}}^{(1)}(x) = P_{\text{LLM}}(x)$, where $x$ represents the samples $(q_i, s, r_i)$, and $P_{\text{LLM}}(x)$ is the distribution of data generated by the LLM $p_L$.

**Proposed Solution** To mitigate the distributional shift in strategy choice between the LLM and the smaller model, we propose to incorporate the smaller model's own correct outputs into the training data. This *self-guided* training with data mixing aligns the training data distribution more closely with the smaller model's output distribution, making learning more effective. A visualization of the data mixing approach is presented in Figure 2 that demonstrates that data mixing reduces the distribution shift, bringing the LLM and the smaller model's output distribution closer. This allows the smaller model to choose the right strategy for a given task, much like the LLM.

## 3 SIKED: SELF-GUIDED ITERATIVE KNOWLEDGE DISTILLATION

We propose SIKeD, an *iterative* training where smaller models can take advantage of their own generations to refine their strategy choices for a given task. In a nutshell, we generate data from the smaller model, filter out the correct samples based on whether the generated solutions are correct, and mix this data with the LLM-generated data to adjust its strategy preferences. The smaller distilled model is used to iteratively generate data in an on-policy setting where it updates itself by leveraging both the LLM data and its own generations. This iterative process allows the smaller model to improve its reasoning abilities and strategy selection over time by leveraging the LLM's knowledge and its own prior learning. The following paragraphs discuss the steps involved in our proposed iterative distillation methodology and the training objective.

**Data generation** For each question $q_i$ and its associated reasoning strategy $s$, we first generate $K$ rationales using the current smaller model $p_{\text{sm}}^\theta$ as: $r_i^{(k)} \sim p_{\text{sm}}^\theta(\cdot \mid s, q_i, I)$, for $k = 1, \ldots, K$. Note that we generate multiple samples $K$ as the likelihood of a correct answer being present in one of the

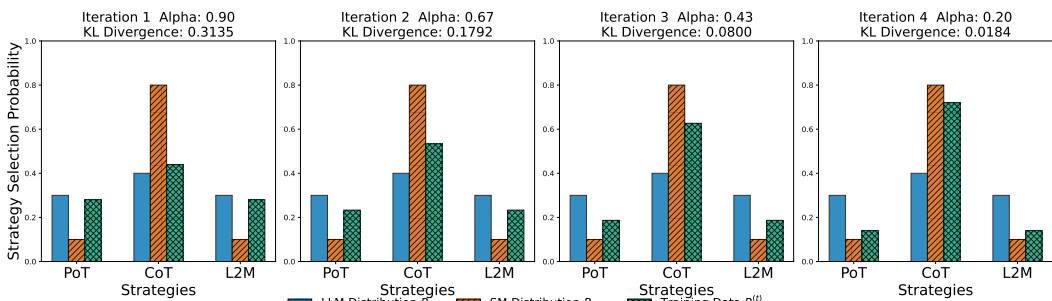

Figure 2: **Alignment of the smaller model's strategy distribution with the LLM over iterations**. Each subplot represents an iteration in the training process, showing the probability distributions over reasoning strategies: PoT, L2M, and CoT. The blue bars depict the LLM's distribution $P_L$, while the orange bars represent the smaller model's distribution $P_{SM}$, which is biased towards CoT. The green bars show the training data distribution $P_{\text{train}}^{(t)}$, a mixture of $P_L$ and $P_{SM}$ weighted by the mixing rate $\alpha$. As $\alpha$ decreases over iterations (from 0.90 to 0.20), $P_{\text{train}}^{(t)}$ shifts from being similar to the LLM's distribution towards the smaller model's distribution. The KL divergence between the training data and the smaller model distributions decreases accordingly, indicating increased similarity.

rationales increases significantly with additional generations for smaller models (Jain & Shridhar, 2023; Wang et al., 2023).

**Data Filtering**   Next, we extract the predicted answer $\hat{a}_i^{(k)}$ from each rationale $r_i^{(k)}$ and compare it with the ground truth $a_i$. We collect the correct samples, where $\hat{a}_i^{(k)} = a_i$, into a new dataset $\mathcal{D}_{\texttt{self}} = \left\{ (q_i, s, r_i^{(k)}) \,\middle|\, \hat{a}_i^{(k)} = a_i \right\}$.

**Data mixing**   We combine the LLM-generated dataset $\mathcal{D}_{\texttt{LLM}}$ with the self-generated dataset $\mathcal{D}_{\texttt{self}}$ to form the mixed dataset $\mathcal{D}_{\texttt{mix}} = \mathcal{D}_{\texttt{LLM}} \cup \mathcal{D}_{\texttt{self}}$.

Note that, we do not always use all the data from LLM in $\mathcal{D}_{\texttt{mix}}$, and study two variations: *All* when all LLM data is used in $\mathcal{D}_{\texttt{mix}}$, and *Sparse* when only queries that have no correct generations in $\mathcal{D}_{\texttt{self}}$ are taken from $\mathcal{D}_{\texttt{LLM}}$. *Sparse* uses less generated data from the LLM, resulting in more computationally efficient training.

The corresponding training data distribution changes to a mixture of the LLM data distribution and the model's own output distribution:

$$P_{\text{train}}^{(2)}(x) = \alpha P_{\text{LLM}}(x) + (1 - \alpha) P_{\text{sm}}^{\theta}(x),$$

where $\alpha = \frac{|\mathcal{D}_{\texttt{LLM}}|}{|\mathcal{D}_{\texttt{LLM}}| + |\mathcal{D}_{\texttt{self}}|}$ serves as a normalized mixing rate between the two datasets.

**Training objective**   By including $\mathcal{D}_{\texttt{self}}$ in the training data, we reduce the divergence between $P_{\text{train}}^{(2)}(x)$ and the model's own output distribution $P_{\text{sm}}^{\theta}(x)$, thus minimizing the distribution shift and improving training effectiveness of choosing the right strategy for a given task.

We continue training the smaller model on $\mathcal{D}_{\texttt{mix}}$ using the following loss function:

$$\mathcal{L}_{\texttt{mix}}(\theta) = -\mathbb{E}_{(q_i, s, r_i) \sim \mathcal{D}_{\texttt{mix}}} \left[ \log p_{\text{sm}}^{\theta}\left( s \mid q_i, I \right) + \sum_{t=1}^{M} \log p_{\text{sm}}^{\theta}\left( r_{i,t} \mid r_{i,<t}, s, q_i, I \right) \right] \quad (2)$$

The expected loss over the training data is:

$$\mathcal{L}_{\texttt{mix}}(\theta) = -\mathbb{E}_{x \sim P_{\text{train}}^{(2)}(x)} \left[ \log p_{\text{sm}}^{\theta}(x) \right]$$

where $x = (q_i, s, r_i)$, and $p_{\text{sm}}^\theta(x)$ denotes the probability assigned by the model to the sample $x$.

**Analogous to minimizing the KL divergence**   Mixing the data is analogous to minimizing the Kullback-Leibler (KL) divergence Kullback & Leibler (1951) between the training data distribution $P_{\text{train}}^{(2)}(x)$ and the model's output distribution $P_{\text{sm}}^\theta(x)$:

$$D_{\text{KL}}(P_{\text{train}}^{(2)}(x) \parallel P_{\text{sm}}^\theta(x)) = \sum_x P_{\text{train}}^{(2)}(x) \log \frac{P_{\text{train}}^{(2)}(x)}{P_{\text{sm}}^\theta(x)}.$$

As we include more self-generated data, $(1 - \alpha)$ increases, and $P_{\text{train}}^{(2)}(x)$ becomes closer to $P_{\text{sm}}^\theta(x)$. This reduces the KL divergence and aligns the training data distribution with the model output distribution, leading to more effective learning. Figure 2 demonstrates that as the value of $\alpha$ decreases, the distribution of the training data strategy choices aligns with the smaller model, with a reduction in their KL value over iterations. This allows the smaller model to better capture the strategy distribution of the larger model.

## 3.1 ITERATIVE SELF-TRAINING OF SIKED

We repeat the data generation, filtering, mixing, and training steps iteratively. In each iteration $t$, the smaller model potentially generates new correct rationales that are added to the training data. The training data distribution at iteration $t$ becomes:

$$P_{\text{train}}^{(t)}(x) = \alpha^{(t)} P_{\text{LLM}}(x) + (1 - \alpha^{(t)}) P_{\text{sm}}^{\theta^{(t-1)}}(x),$$

where $\theta^{(t-1)}$ are the model parameters from the previous iteration, and $\alpha^{(t)}$ is updated based on the sizes of $\mathcal{D}_{\text{LLM}}$ and $\mathcal{D}_{\text{self}}^{(t)}$ at iteration $t$. Note that the generated samples from the smaller model automatically govern the value of $\alpha^{(t)}$ based on the size of $\mathcal{D}_{\text{self}}^{(t)}$.

This iterative process continues until the model's performance converges or a predefined stopping criterion is met. Over multiple iterations, the model's own output distribution $P_{\text{sm}}^{\theta^{(t)}}(x)$ gradually improves, and the training data distribution becomes increasingly aligned with it. We present an end-to-end training methodology in Algorithm 1.

## 4 EXPERIMENTAL DETAILS

**Dataset**   Our work demonstrates the effectiveness of selecting an appropriate strategy for a given task. We consider multi-step mathematical reasoning datasets in our work, as various strategies can solve the task fairly well. We trained SIKeD on the GSM8K training set Cobbe et al. (2021), which includes 7,473 samples, and tested it on the corresponding test set of 1,319 samples. To assess the domain transferability of our distillation method, we also evaluated it on three additional mathematical datasets: SVAMP Patel et al. (2021) with 1,000 samples, ASDiv Miao et al. (2020) with 2,300 test samples, and MultiArith Roy & Roth (2016) with 180 samples. As the GSM8K training set was used to train the smaller model, we classify it as an *in-distribution* dataset. In contrast, no training data from SVAMP, ASDiv, or MultiArith was used, as they were exclusively employed for testing purposes and thus considered *out-of-distribution*.

**Implementation Details**   We used the Llama3 70B model Dubey et al. (2024) as the large language model (LLM) to generate the rationales. We performed distillation on different smaller models ranging from 0.5B to 7B parameters, including Qwen2 0.5B Bai et al. (2023), Qwen2 1.5B Bai et al. (2023), SmolLM 1.7B Hugging Face (2023), Gemma 2B Team et al. (2024), and Gemma 7B Team et al. (2024). All smaller models were fine-tuned using LoRA Hu et al. (2021) with a rank of 16, and alpha of 32. We used a learning rate of 3e-4 for Qwen models with a cyclic scheduler, while we set 2e-4 as the learning rate for other models and used a linear scheduler. We train all models for 3 epochs. We implemented all our experiments using the Unsloth FastLanguageModel Unslothai (2023) and used the VLLM library Kwon et al. (2023) for inference. We set the temperature $t = 0$ for data generation from the LLM while $t = 0.7$ was used for generating samples from the smaller model

---

**Algorithm 1: SIKeD: Self-guided Iterative Knowledge Distillation**

---

**Input:** $\mathcal{D}$: Reasoning dataset with questions $\{q_i\}_{i=1}^N$ and answers $\{a_i\}_{i=1}^N$, $\mathcal{D}_{\text{LLM}}$: Reasoning dataset generated using the LLM with questions $\{q_i\}$, answers $\{a_i\}$, strategy $\{s\}$, rationales $\{r_i\}$, $S$: Set of reasoning strategies, $I$ : instruction, $p_L$: LLM for rationale generation , $p_{\text{sm}}^{\theta^{(0)}}$: Smaller model with initial parameters $\theta^{(0)}$ , $K$: Number of samples per question and strategy , $T$: Maximum number of iterations , **Variation**: *All* or *Sparse*,

// LLM-Based Distillation

Train $p_{\text{sm}}^{\theta^{(0)}}$ on $\mathcal{D}_{\text{LLM}}$ by minimizing $\mathcal{L}_{\text{L}}(\theta^{(0)})$ ( equation 1)

// SIKeD: **S**elf-guided **I**terative **K**nowledge **D**istillation

**for** *iteration $t = 1$ to $T$* **do**

    **Initialize** dataset $\mathcal{D}_{\text{self}}^{(t)} \leftarrow \emptyset$

    **for** *each question $q_i \in \mathcal{D}$* **do**

        **for** *each strategy $s \in S$* **do**

            **for** $k = 1$ to $K$ **do**

                Generate rationale $r_i^{(k)}$ using $p_{\text{sm}}^{\theta^{(t-1)}}$: $r_i^{(k)} \sim p_{\text{sm}}^{\theta^{(t-1)}}(\cdot \mid s, q_i, I)$

                Extract answer $\hat{a}_i^{(k)}$ from $r_i^{(k)}$

                **if** $\hat{a}_i^{(k)} = a_i$ **then**

                    Add $(q_i, s, r_i^{(k)})$ to $\mathcal{D}_{\text{self}}^{(t)}$

                **end**

            **end**

        **end**

    **end**

    **if** *Variation is* All **then**

        Combine datasets: $\mathcal{D}_{\text{mix}}^{(t)} = \mathcal{D}_{\text{LLM}} \cup \mathcal{D}_{\text{self}}^{(t)}$

    **else**

        Identify questions with no correct self-generated rationales: $\mathcal{I} = \{i \mid \text{no correct } r_i^{(k)} \text{ in } \mathcal{D}_{\text{self}}^{(t)}\}$

        Include corresponding LLM data: $\mathcal{D}_{\text{LLM}}^{(t)} = \{(q_i, s, r_i) \in \mathcal{D}_{\text{LLM}} \mid i \in \mathcal{I}\}$

        Combine datasets: $\mathcal{D}_{\text{mix}}^{(t)} = \mathcal{D}_{\text{LLM}}^{(t)} \cup \mathcal{D}_{\text{self}}^{(t)}$

    **end**

    Update $\alpha^{(t)} = \frac{|\mathcal{D}_{\text{LLM}}^{(t)}|}{|\mathcal{D}_{\text{LLM}}^{(t)}| + |\mathcal{D}_{\text{self}}^{(t)}|}$

    Retrain $p_{\text{sm}}^{\theta^{(t)}}$ on $\mathcal{D}_{\text{mix}}^{(t)}$ by minimizing $\mathcal{L}_{\text{mix}}^{(t)}(\theta^{(t)})$ ( equation 2)

**end**

**Output:** Updated smaller model $p_{\text{sm}}^{\theta^{(T)}}$

---

at each iteration. We set the number of generated samples or $K$ to 10. We report Top-1 accuracy (maj@1).

Our proposed approach was compared against a set of single-strategy distillation methods. In this work, we employed three reasoning strategies: Chain-of-Thought (CoT) Wei et al. (2022), Program-of-Thought (PoT) Chen et al. (2023), and Least-to-Most (L2M) Zhou et al. (2023).

## 5 RESULTS AND DISCUSSION

**LLM Based Distillation** We start by distilling smaller models using the reasoning dataset generated using the LLM in two variations: using data from a single strategy (CoT, PoT, or L2M), and a combination of all three strategies (referred to as "Combined"). Table 1 compares the accuracies of the approaches across four mathematical datasets. The "Combined" approach benefited smaller models, yielding slight improvements for the Qwen 0.5B, Qwen 1.5B, and SmolLM 1.7B models. However, it showed little to no improvement, and sometimes even worse performance, for the larger Gemma 2B and 7B models. This indicates that simply merging the distillation data for each strategy is not sufficient for effective multi-strategy distillation.

**Consistent improvement across in-distribution dataset** Compared to the traditional LLM-based distillation approaches, we observe consistent improvements with SIKeD across all models, ranging from 0.5B to 7B parameters as shown in Table 1. On the in-distribution GSM8K dataset, both Gemma

Table 1: Top-1 (maj@1) accuracy comparison across different sized models (Gemma 2B and 7B, SmolLM 1.7B, Qwen 0.5B and 1.5B) on four mathematical datasets: GSM8K, ASDiv, MultiArith, and SVAMP. "Combined" refers to the scenario where data from all three reasoning strategies are merged and then used for distillation. We report the two variants of `SIKeD`: "Sparse" and "All".

| Model | Method | In Distribution | Out of Distribution | | |
|---|---|---|---|---|---|
| | | GSM8K | ASDiv | MultiArith | SVAMP |
| Gemma 7B | CoT | 67.40 | 68.76 | 98.33 | 66.80 |
| | L2M | 69.29 | 64.69 | 96.11 | 64.80 |
| | PoT | 71.34 | 67.85 | 98.89 | 75.00 |
| | Combined | 70.74 | 69.11 | 99.44 | 69.40 |
| | SIKeD | | | | |
| | Sparse | **73.84** (↑ +2.5) | **70.59** (↑ +1.5) | 99.44 (-) | 72.90 (↓ -2.1) |
| | All | 72.55 (↑ +1.2) | 70.33 (↑ +1.2) | **100.0** (↑ +0.6) | **75.00** (↑ +0.0) |
| Gemma 2B | CoT | 36.54 | 54.01 | 87.22 | 41.90 |
| | L2M | 36.92 | 43.47 | 81.67 | 31.60 |
| | PoT | 44.05 | 58.13 | 90.56 | 56.80 |
| | Combined | 44.05 | 57.96 | 84.44 | 56.20 |
| | SIKeD | | | | |
| | Sparse | **47.23** (↑ +3.2) | 59.05 (↑ +0.8) | **91.11** (↑ +0.6) | **58.60** (↑ +1.8) |
| | All | 46.02 (↑ +2.0) | **59.39** (↑ +1.3) | **91.11** (↑ +0.6) | 57.50 (↑ +0.7) |
| SmolLM 1.7B | CoT | 16.38 | 30.37 | 58.89 | 22.60 |
| | L2M | 18.73 | 22.13 | 53.89 | 17.90 |
| | PoT | 23.73 | 43.77 | 61.11 | 34.50 |
| | Combined | 24.56 | 46.77 | 67.22 | 35.90 |
| | SIKeD | | | | |
| | Sparse | **27.98** (↑ +3.4) | **47.20** (↑ +0.4) | **72.22** (↑ +5.0) | **37.80** (↑ +1.9) |
| | All | 26.54 (↑ +2.0) | 45.47 (↓ -1.3) | 70.56 (↑ +3.2) | 36.30 (↑ +0.4) |
| Qwen 1.5B | CoT | 55.57 | 68.76 | 99.44 | 66.30 |
| | L2M | 54.59 | 63.69 | 96.67 | 62.30 |
| | PoT | 64.22 | 66.94 | 95.56 | 74.30 |
| | Combined | 64.44 | 67.64 | 98.89 | 73.20 |
| | SIKeD | | | | |
| | Sparse | **64.97** (↑ +0.5) | **68.98** (↑ +1.3) | **99.44** (-) | **75.40** (↑ +1.1) |
| | All | 64.14 (↓ -0.3) | 67.72 (↑ +0.1) | **99.44** (-) | 74.50 (↑ +0.2) |
| Qwen 0.5B | CoT | 36.47 | 54.66 | 83.89 | 43.00 |
| | L2M | 33.59 | 49.76 | 76.67 | 44.60 |
| | PoT | 41.62 | 56.83 | 92.22 | 51.40 |
| | Combined | 42.38 | 57.79 | 90.56 | 51.40 |
| | SIKeD | | | | |
| | Sparse | 43.14 (↑ +0.8) | **58.44** (↑ +0.7) | 93.33 (↑ +1.1) | **51.70** (↑ +0.3) |
| | All | **44.28** (↑ +1.9) | 58.05 (↑ +0.3) | **95.00** (↑ +2.8) | **51.70** (↑ +0.3) |

2B and 7B show significant gains of +3.2 points and +2.5 points respectively (44.05 → 47.23 and 71.34 → 73.84, respectively). Similarly, SmolLM showed the largest improvement of +3.4 points (24.56 → 27.98). In contrast, the smaller Qwen models see gains of +0.5 points for the larger variant (1.5B) and +1.9 points for the smaller variant (0.5B).

**`SIKeD` performs well on out-of-distribution datasets** For the out-of-distribution datasets, there is a steady improvement on the ASDiv dataset, with Gemma 7B gaining +1.5 points (69.11 → 70.59), +1.3 points for Gemma 2B (58.13 → 59.39), +0.4 points for SmolLM, +1.3 points for Qwen 1.5B (67.64 → 68.98), and +0.7 points for Qwen 0.5B (57.79 → 58.44). A similar trend is seen for the MultiArith dataset, where SmolLM shows the largest gain of +5 points. It is followed by Qwen 0.5B with +2.8 points, while other models outperform the baseline. In particular, Gemma 7B achieves a perfect score of 100. The results are similar for the SVAMP dataset, with Qwen 0.5B, Qwen 1.5B, SmolLM 1.7B, and Gemma 2B gaining +0.3, +1.1, +1.9, and +1.8 points, respectively. On the other hand, Gemma 7B maintains its baseline score of 75. Upon further analysis, we find that the SVAMP dataset tends to favor the PoT strategy, which outperforms other strategies by up to 10 points for

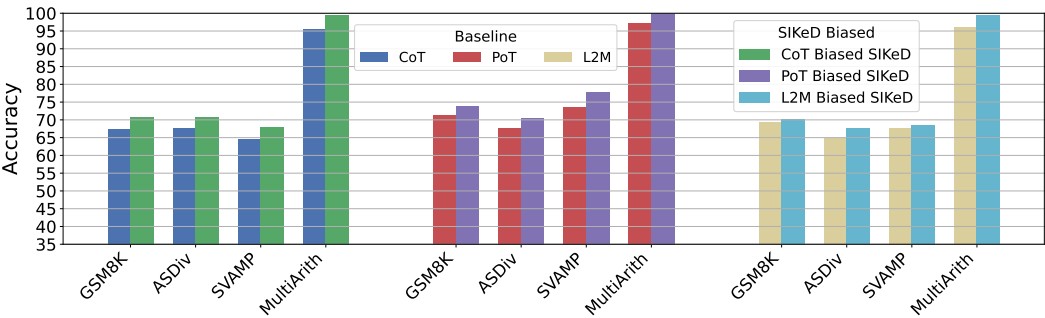

Figure 3: Accuracy comparison between single distillation strategies of CoT, PoT, and L2M with `SIKeD` biased training using the same strategy using the Gemma 7B model.

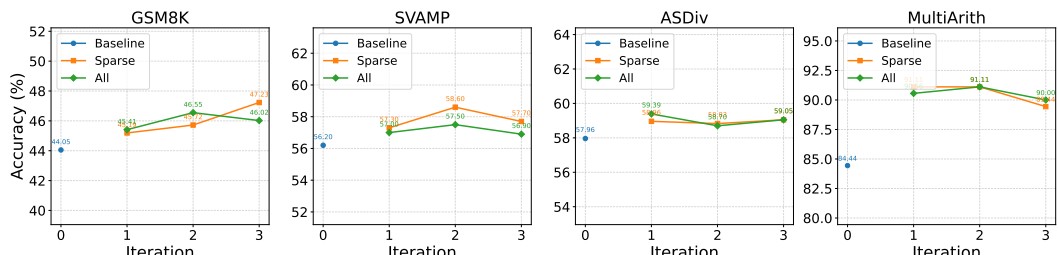

Figure 4: Iterative accuracy comparison for the Gemma 2B model across all datasets. The process is stopped when the gains diminish or when it is no longer cost effective to continue.

Gemma 7B. Although our approach often selects PoT, it does not always do so, leading to results similar to the baseline score.

**Biasing `SIKeD` in favor of our strategy of choice**    For some tasks, one strategy might be better than the others due to its performance, lower cost, or better suitability for some use cases (for example, PoT is significantly better for SVAMP compared to other strategies). In such cases, it would be beneficial to bias the selection towards that strategy [3]. This can be done by taking only the sample from our choice of biased strategy when more than one strategy is correct from the model-generated samples. For example, if for a given data point, a smaller model samples both CoT and PoT correctly, and our biased strategy choice is PoT, we will ignore the CoT output and take only the PoT. Figure 3 compares the individual distillation strategy with the biased `SIKeD`. Using Gemma 7B as a smaller model across all datasets, `SIKeD` outperforms individual distillation strategies by a margin of 2-4 points, highlighting the effectiveness of `SIKeD` over other distillation approaches.

**How many iterations to run for `SIKeD`**    With each iteration of `SIKeD`, the model learns to solve a task using different strategies and adjusts its strategy choice for a given task. This allows for continuous training of `SIKeD`. Figure 4 illustrates the accuracy improvements across iterations for the Gemma 2B model on various datasets. The iterative training is stopped when accuracy shows only marginal improvements or declines. Three iterations have consistently proven to be the optimal balance across different models and datasets in our experiments.

**How the strategy distribution changes over iterations**    Figure 5 illustrates the strategy distribution across different iterations for the GSM8K dataset using the SmolLM 1.7B model. Iteration 0 represents the baseline "combined" training from Table 1, and as expected, the smaller model is initially biased towards one strategy (PoT in this case). Iterations 1, 2, and 3 show the model's progression using `SIKeD`, where it learns to diversify and select the suitable strategy for the given problem. Notably, while PoT remains the dominant strategy, the model improves its usage of the

---

[3]Note that this is different from the already biased selection of the smaller model, as our biased strategy may not be the default biased choice of the smaller model.

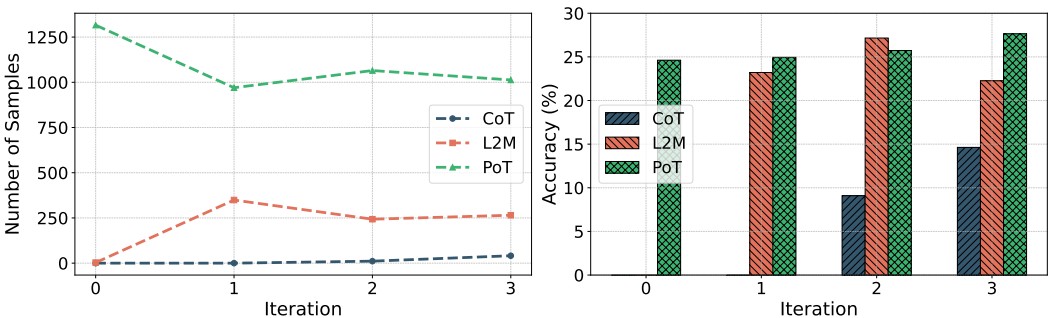

Figure 5: Strategy distribution over iterations for GSM8K dataset using SmolLM 1.7B model.

other two strategies—CoT and L2M—which were absent in the biased baseline. This diversification of strategies results in an overall gain of +3 points over the baseline.

**Training from the last checkpoint vs training from pre-trained checkpoint**  In our work, we iteratively train from the last checkpoint using on-policy training as we expect continuous improvements in the model performance with a newly learned strategy. However, an alternative approach uses off-policy training (training the pre-trained model at each iteration) to achieve strong performance Gulcehre et al. (2023). We compared on-policy training (our proposed approach) with off-policy training (as in Gulcehre et al. (2023)), utilizing both LLM-generated and self-generated data, and observed a notable decrease in the overall accuracy with off-policy training. Note that we used all of the LLM data at each iteration for off-policy training as the training is done on the pre-trained model. On the GSM8K dataset, our on-policy approach outperformed off-policy training by +6 points (45.19 vs 38.90) using the Gemma 2B model. A similar trend was observed on out-of-distribution datasets, where `SIKeD` outperformed with a gain of +4 points on both the SVAMP and ASDiv datasets (55.4 vs 51.3 for SVAMP and 58.96 vs 55.44 for ASDiv) and a gain of +2 points on the MultiArith dataset (91.11 vs 88.33).

**Is data mixing better than LLM-based distillation or self-distillation**  Mixing data from the smaller model with the LLM helps in bridging the distributional gap between the LLM and the smaller model. We explore the role of the data mixing rate alpha ($\alpha$) to validate our hypothesis regarding data mixing. When $\alpha$ is set to 1, only LLM-generated data is used, while at the other extreme, $\alpha = 0$ means only data generated by the smaller model is used. As shown in Figure 6, an $\alpha$ value between 0 and 1 performs better than either of the extreme cases.

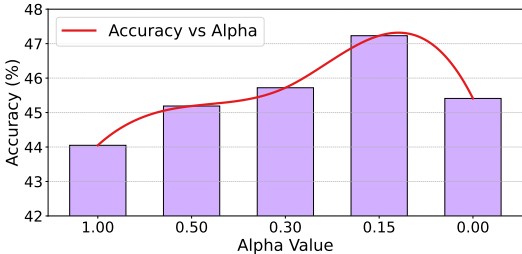

Figure 6: Accuracy comparison with different data mix controlled by $\alpha$. $\alpha = 1$ means all data is from the LLM only, while $\alpha = 0$ means only smaller model data is used.

**Qualitative analysis demonstrating that smaller models learn to choose the right strategy over iterations**  The increase in the performance of smaller models can be attributed to the change in the distribution of reasoning strategies as the model becomes more aligned with its strategy choices. Figure 7 shows that a model can solve a given problem in the third iteration by switching the reasoning strategy which it initially struggled with. Furthermore, Figure 11 shows an example of a model correcting its reasoning chain over time due to its improved reasoning capabilities.

## 6  RELATED WORK

**Knowledge Distillation for Reasoning Tasks**  Knowledge distillation (Buciluǎ et al., 2006; Hinton, 2015) is a widely-used technique for transferring knowledge from a large language model (LLM) to a smaller model. Previous research has focused on transferring intermediate reasoning steps to

**Question** There are twice as many boys as girls at Dr. Wertz's school. If there are 60 girls and 5 students to every teacher, how many teachers are there?

**Iteration2**
girls = 60
boys = 2 ∗ girls
total_students = girls + boys
teachers_per_student = 5
total_teachers = total_students * teachers_per_student
answer = total_teachers
Final Answer: 900 ✗

**Baseline**
girls = 60
boys = 2 ∗ girls
teachers = girls/5
answer = teachers
Final Answer: 12 ✗

**Iteration1**
girls = 60
boys = 2 ∗ girls
teachers = girls/5
answer = teachers
Final Answer: 12 ✗

**Iteration3**
Sub-question 1: How many boys are there?
Answer to Sub-question 1: $2 * 60 = << 2 * 60 = 120 >> 120$
Sub-question 2: How many students are there in total?
Answer to Sub-question 2: $60 + 120 = << 60 + 120 = 180 >> 180$
Sub-question 3: How many teachers are there?
Answer to Sub-question 3: $180/5 = << 180/5 = 36 >> 36$
Final Answer: 36 ✓

Figure 7: A qualitative example showing how Gemma 2B shifts its strategy selection over iterations to solve a given task.

smaller models, either step-by-step (Shridhar et al., 2023; Magister et al., 2023; Hsieh et al., 2023) or by combining multiple strategies simultaneously Chenglin et al. (2023); Zhu et al. (2024); Li et al. (2024). These approaches can be viewed as aggregating diverse data sources for distillation, similar to the LLM data approach in our work. In contrast, Hahn & Choi (2019) and Xu et al. (2020) focus on *self-distillation*, where a model improves its performance without external data or knowledge. Specifically, Hahn & Choi (2019) leverages word embeddings, while Xu et al. (2020) uses temporal model checkpoints as a proxy for ground truth. However, both approaches rely solely on data generated by the smaller model and exclude LLM data. Our method strikes a balance between these two extremes by using LLM data to learn multiple strategies and self-generated data to optimize for the right strategy choice.

**Self-learning** Previous studies, such as (He et al., 2019; Sun et al., 2020; Gulcehre et al., 2023), have shown the effectiveness of the *self-training paradigm* in tasks such as machine translation. While ReST Gulcehre et al. (2023) uses off-policy training, we find on-policy training more suitable for our case both in terms of data efficieny and performance. On-policy training also allows a better choice of learning strategies, since the model can use its most recent learning. Agarwal et al. (2024) introduces Generalized Knowledge Distillation (GKD), an on-policy training method that aligns the distributions of large language models (LLMs) and smaller models by incorporating output sequences sampled from the student during training. However, the task was limited to the distribution alignment and not to aligning the strategy choices in a multi-strategy distillation. Simply applying GKD would not address this issue, as it would force the smaller model to learn all strategies, which is impractical given its limited capacity.

Finally, we compare our distillation strategies with LLM-based distillation using both individual strategies Shridhar et al. (2023); Magister et al. (2023); Hsieh et al. (2023) and a combination of several strategies at once Chenglin et al. (2023); Zhu et al. (2024).

## 7 CONCLUSION

We propose `SIKeD`: **S**elf-guided **I**terative **K**nowledge **D**istillation, that addresses the challenge of distilling multistep reasoning skills from large language models (LLMs) to smaller models. Unlike traditional methods, which often leave smaller models biased towards a single strategy, `SIKeD` uses iterative *self-guided* training, combining LLM and self-generated data to improve overall reasoning in smaller models. We demonstrate our approach across various mathematical datasets and demonstrate that `SIKeD` improves the ability of smaller models to handle complex reasoning tasks, achieving significant performance gains.

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

# A APPENDIX

---

Below is an instruction that describes a task, paired with an input that provides further context. Write a response that appropriately completes the request.

Instruction: Solve the given math problem step by step. Put your final answer after 'Final answer:'.

Input: John buys 3 dress shirts. They sell for $20 each. He also has to pay 10% tax on everything. How much did he pay in total?

Response: The shirts cost 3*$20=$<<3*20=60>>60 before tax The tax cost $60*.1=$<<60*.1=6 6 So in total they paid $60+$6=$<<60+6=66 66 Final Answer: 66<eos>

**[7 more examples randomly sampled from the training set]**

Input: Thomas is training at the gym to prepare for a competition. He trained for 5 hours every day for a month (30 days). If he continues to train for the next 12 days, how many hours will he spend on training in total?

Response: Model generated response ..

---

Figure 8: 8-shot Chain of Thought demonstration.

---

Below is an instruction that describes a task, paired with an input that provides further context. Write a response that appropriately completes the request.

Instruction: Solve the given math problem by decomposing it into smaller, manageable sub-questions. Put your final answer after 'Final answer: '.

Input: John buys 3 dress shirts. They sell for $20 each. He also has to pay 10% tax on everything. How much did he pay in total?

Response: Sub-question 1: How much did John spend on shirts?
Answer to Sub-question 1: 3*$20=$<<3*20=60>>60
Sub-question 2: How much did John spend on tax?
Answer to Sub-question 2: $60*.1=$¡¡60*.1=6¿¿6
Sub-question 3: How much did John spend in total?
Answer to Sub-question 3: $60+$6=$<<60+6=66>>66
Final Answer: 66<eos>

**[7 more examples randomly sampled from the training set]**

Input: Thomas is training at the gym to prepare for a competition. He trained for 5 hours every day for a month (30 days). If he continues to train for the next 12 days, how many hours will he spend on training in total?

Response: Model generated response ..

---

Figure 9: 8-shot Least to Most demonstration.

---

Below is an instruction that describes a task, paired with an input that provides further context. Write a response that appropriately completes the request.

Instruction: Solve the given math problem by writing a python program. Store your result as a variable named 'answer'.

Input: John buys 3 dress shirts. They sell for $20 each. He also has to pay 10% tax on everything. How much did he pay in total?

Response: total_shirts = 3
cost_of_one_shirt = 20
total_cost_shirts = total_shirts * cost_of_one_shirt
tax_rate = 0.1
tax_amount = tax_rate * total_cost_shirts
total_cost = total_cost_shirts + tax_amount
answer = total_cost <eos>

**[7 more examples randomly sampled from the training set]**

Input: Thomas is training at the gym to prepare for a competition. He trained for 5 hours every day for a month (30 days). If he continues to train for the next 12 days, how many hours will he spend on training in total?

Response: Model generated response ..

---

Figure 10: 8-shot Program of Thought demonstration.

---

**Question** In a dance class of 20 students, 20% enrolled in contemporary dance, 25% of the remaining enrolled in jazz dance, and the rest enrolled in hip-hop dance. What percentage of the entire students enrolled in hip-hop dance?

**Baseline**
total_students = 20
percentage_enrolled_contemporary = 0.2
students_enrolled_contemporary = total_students * percentage_enrolled_contemporary
remaining_students = total_students − students_enrolled_contemporary
percentage_enrolled_jazz = 0.25
students_enrolled_jazz = remaining_students * percentage_enrolled_jazz
remaining_students = remaining_students - students_enrolled_jazz
percentage_enrolled_hip_hop = 1 − percentage_enrolled_contemporary − percentage_enrolled_jazz
answer = percentage_enrolled_hip_hop
Final Answer: 0.55 ✗

**Iteration1**
total_students = 20
contemporary_students = total_students * 0.2
remaining_students = total_students − contemporary_students
jazz_students = remaining_students * 0.25
hip_hop_students = remaining_students − jazz_students
percentage_hip_hop = hip_hop_students/total_students * 100
answer = percentage_hip_hop
Final Answer: 60 ✓

---

Figure 11: Over iteration, Gemma 2B model learnt to fix its mistake using the same strategy

