# OpenReview forum: "SIKeD: Self-guided Iterative Knowledge Distillation for Mathematical Reasoning"
_ICLR.cc/2025/Conference — Submitted to ICLR 2025_

### Official Review · Reviewer_ytP5 · 2024-10-27

**Soundness:** 3
**Presentation:** 4
**Contribution:** 2
**Rating:** 5
**Confidence:** 4

**Summary:**

This paper addresses the challenge of knowledge transfer from large language models (LLMs) to smaller, more efficient models. The authors identify that smaller models typically demonstrate constrained capabilities and tend to favor single reasoning strategies. To address this limitation, the authors introduce SIKED, an approach that leverages self-generated outputs to create iteratively mixed training datasets. The method promotes the development of diverse reasoning strategies during the knowledge transfer process. The effectiveness of SIKED is demonstrated through empirical evaluations on several mathematical reasoning datasets, showing improvements over baseline approaches.

**Strengths:**

The proposed problem is worth exploring.
The paper is well-written and easy to follow.

**Weaknesses:**

1. Although the method demonstrates effectiveness, the motivation requires further clarification. While iteratively forming new training datasets based on model outputs is an existing approach, the paper's contribution lies in showing this method can enable diverse reasoning strategies in smaller models. However, the underlying mechanism for how this approach promotes strategy diversity needs clearer explanation.
2. The experimental evaluation is currently limited to mathematical reasoning tasks. Exploring the effectiveness of the proposed method in other scenarios would provide valuable insights into its generalization capabilities.

**Questions:**

1. Can the authors elaborate on the underlying reasons for smaller models' bias towards specific strategies? Does the preferred strategy vary?
2. The paper states that mixing LLM-generated data with self-generated outputs helps align smaller models with their learned knowledge (L83). However, discarding samples with mismatched outputs creates a different form of bias? Please clarify this.
3. Please explain the mechanism by which mixing the data promotes diverse strategy selection in smaller models.
4. How to judge if the generated r_i is correct? (L294)
5. Is it a typo in L211 & L215 that the same notation appears in different contexts?
6. Could the authors clarify the strategy sampling process for smaller models? Specifically regarding L413, "if both CoT and PoT are sampled correctly, our biased strategy choice is PoT" - is this strategy determined by the smaller model's output? Additional details on the strategy sampling mechanism would be appreciated and helpful

---

> ### Author Response · Authors · 2024-11-15
> **Addressing weakness pointed by the reviewers**
>
> We thank the reviewer for a careful examination of our work, and providing positive feedback for our research problem. We discuss the weaknesses and questions highlighted by the reviewer below.
>
> ## Weaknesses
> ### How SIKeD promotes strategy diversity
> >Although the method demonstrates effectiveness, the motivation requires further clarification. While iteratively forming new training datasets based on model outputs is an existing approach, the paper's contribution lies in showing this method can enable diverse reasoning strategies in smaller models. However, the underlying mechanism for how this approach promotes strategy diversity needs clearer explanation.
>
> Knowledge distillation uses all data from the LLMs while self-distillation relies on smaller models data for training. SIKeD lies in the middle of the two extremes, where at each iteration, the smaller model generates multiple outputs (k=10) and a reward is applied to it (0 if incorrect, 1 if correct). **The correct reward incentivizes the model to improve its performance over iterations and also to choose the right strategy for a given task which may not be the greedy strategy making the strategy choice more diverse** (if needed). In a sense, optimizing for performance automatically leads to strategy diversity over iterations, since the smaller model cannot solve all the questions with the same strategy. **The qualitative analysis presented in Fig. 7 shows that the smaller model changes its strategy to get the correct solution if needed**. Essentially, in our setting, diversifying strategy is entangled with improved performance.
>
> ### The experimental evaluation is currently limited to mathematical reasoning tasks. Exploring the effectiveness of the proposed method in other scenarios would provide valuable insights into its generalization capabilities.
>
> We have discussed this in detail in the general comments section. In short, we wanted to demonstrate the self-iterative knowledge distillation approach where a smaller model can learn to pick the right strategy to solve a given task following LLM. We used Chain of Thought, subquestion decomposition and Program of Thought as our strategies to pick from. Essentially **we were looking for a task that possesses the qualities of intermediate reasoning, can be decomposed into smaller tasks and can be represented in the form of a program. We found mathematical reasoning as the perfect fit for our use case** and other reasoning tasks were hard to fit into the chosen strategies. This is why we limited ourselves to mathematical reasoning and also added that to the title to not oversell our idea. We are collecting a dataset in a different domain that can work across multiple strategies but that is for future work.

---

> ### Author Response · Authors · 2024-11-15
> **Addressing Reviewers questions**
>
> Below is a discussion on the questions raised by the reviewer.
>
> ### Q1: Can the authors elaborate on the underlying reasons for smaller models' bias towards specific strategies? Does the preferred strategy vary?
>
> Our experiments (Fig 5) suggests that **the dominant strategy (measured in terms of the performance) is often preferred by the smaller model**. This might be  because of relatively more samples present in the training data. However, **when we balance the samples per strategy  (for ablation study), the model prefers CoT, possibly because of the pre-training bias**.
>
> ### Q2: Bias introduced due to discarding samples
> >The paper states that mixing LLM-generated data with self-generated outputs helps align smaller models with their learned knowledge (L83). However, discarding samples with mismatched outputs creates a different form of bias? Please clarify this.
>
> Although we discard the mismatched outputs from the smaller model and utilize the correct generations from LLM, **the KL divergence between the training data and SLM still decreases as all the data is not just from the LLM and mixing smaller model data reduces the KL divergence**. This is also demonstrated in Fig. 2.
>
> ### Q3: Please explain the mechanism by which mixing the data promotes diverse strategy selection in smaller models.
>
> - Please refer to Weaknesses 1 above. We have addressed this concern in details.
>
>
> ### Q4: How to judge if the generated r_i is correct? (L294)
> **Knowing if the rationale is correct or not is a strong limitation of all the reasoning tasks involving intermediate steps and is not a specific limitation for our work**. This has been discussed in the past works like https://arxiv.org/pdf/2212.10001
>
> ### Q5: Is it a typo in L211 & L215 that the same notation appears in different contexts?
> L211 is the rolled out version of L215. We did not find any typo in those lines. Please let us know if there is anything specific you would like to point out.
>
> ### Q6: Discussing how selection works for biased strategy
> >Could the authors clarify the strategy sampling process for smaller models? Specifically regarding L413, "if both CoT and PoT are sampled correctly, our biased strategy choice is PoT" - is this strategy determined by the smaller model's output? Additional details on the strategy sampling mechanism would be appreciated and helpful
>
> *The underlying assumption for using a biased strategy selection mechanism lies in the fact that for some datasets, a specific strategy might be more beneficial than the others* (L408-410). Based on this, we create biased training datasets for all the three strategies. **For example, if we are creating a biased dataset for PoT, we firstly select only the problems which can be solved via PoT.** We ignore the CoT and L2M reasoning chains for these problems. For problems that cannot be solved via PoT, we select CoT or L2M chains at an instance level, but not both. This helps us to create a dataset biased on PoT.
>
> Please let us know if all the concerns are addressed. We are happy to discuss more.
> Thank you.

---

> ### Author Response · Authors · 2024-11-18
> **Follow-Up on Rebuttals**
>
> We want to sincerely thank the reviewers for their time and effort in evaluating our paper. We would appreciate it if you could kindly confirm that the rebuttal was received and let us know if any additional steps or clarifications are required from our side. Your feedback is highly important to us, and we remain available to address any further concerns or questions.
>
> Please let us know. Thanks.

---

> > ### Comment · Reviewer_ytP5 · 2024-11-30
> >
> > I appreciate the authors' clarification. My major concern relates to the underlying mechanism of diverse reasoning strategies enabled by adding generated outputs to the training datasets. While this is a major contribution of the proposed method, I find the explanation unclear and lacking comprehensive discussion in the paper. For example, is the generated dataset the source from which the model learns 'diversity'? Additionally, I would be very interested in understanding what specific properties of the generated dataset enable the model to achieve such diversity.

---

> > > ### Author Response · Authors · 2024-12-01
> > >
> > > Dear reviewer ytP5,
> > >
> > > We sincerely thank you for your response. As pointed out in your comment, the generated training dataset is indeed the source from which the model learns diversity. However, there are a few nuances. We detail them below:
> > >
> > > 1. Simply distilling through a diverse dataset does not guarantee the student model to generate diverse outputs. This is shown from the strategy distribution of our baseline 'combined' model (Figure 5 and Line 316-320). Further, it is often the case that student models are unable to generalize across different strategies (Line 068 ; [1] and [2]). To remedy this, SIKeD iteratively generates the training dataset by mixing LLM and SLM (Small Language Model) generated datasets, controlled by the parameter alpha, which in turn depends on the accuracy of SLM predictions. Note that the model learns to improve its diversity in subsequent iterations (Figure 5)
> > >
> > > 2. Regarding the specific properties of the dataset, SIKeD requires that the distribution of the dataset is almost uniform with respect to the different strategies (CoT, PoT, L2M). This is ensured by using an LLM which performs almost equally well with different strategies, in our case LLaMA3 70B model. However, as mentioned in the point above, this is not enough. Although the distribution of the training dataset with respect to the different strategies remains the same over the iterations, SLM + LLM data mixing allows the smaller model to diversify its outputs by iteratively updating the training dataset.
> > >
> > > We plan to update our paper with an appendix section specifically discussing how SIKeD helps the model to diversify its outputs.
> > >
> > > **References**
> > >
> > > [1] Dean A Pomerleau. Efficient training of artificial neural networks for autonomous navigation. Neural computation, 3(1):88–97, 1991.  https://ieeexplore.ieee.org/document/6796843
> > >
> > > [2] Stephane Ross and Drew Bagnell. Efficient reductions for imitation learning. In Yee Whye Teh and Mike Titterington (eds.), Proceedings of the Thirteenth International Conference on Artificial Intelligence and Statistics, volume 9 of Proceedings of Machine Learning Research, pp. 661–668, Chia Laguna Resort, Sardinia, Italy, 13–15 May 2010. PMLR. URL https://proceedings.mlr.press/v9/ross10a.html.

---

### Official Review · Reviewer_YiHw · 2024-10-28

**Soundness:** 2
**Presentation:** 3
**Contribution:** 2
**Rating:** 3
**Confidence:** 3

**Summary:**

This paper proposes **S**elf-guided **I**terative **K**nowledge **D**istillation (SIKeD), which utilizes outputs of LLMs (teacher) and small models (student) to Iterative train the student model. The main process can be summarized as follows:

source data (GSM8K training set) → LLMs generate Cot, PoT and other format reasoning → training small models → small models augment data → filtering → training small models → small models augment data → filtering → … … n …

Source data: GSM8K training set (7,473 samples)

Model: Qwen2 0.5B, 1.5B, SmolLM 1.7B, Gemma 2B / 7B,

**Strengths:**

This paper carefully presents the relevant method analysis and explores how generation strategies evolve across multiple iterations in small models, which is a very intriguing phenomenon.

**Weaknesses:**

1. The mathematical notation is overly verbose. There is no proof or any other theoretical contributions. However, this paper employs overly complex notation. And there is also no revision on loss function. The overly complex notation will prevent quick understanding. The main process of proposed method can be summarized in one or two sentences. And all operations performed at the dataset level.

2. Limited generalization. The approach only enhances the GSM8K dataset, but reasoning tests should be conducted on more realistic datasets, such as MATH, Arc-Challenge and so on. And more reasoning tasks also need to be evaluated, such as commonsense reasoning and symbolic reasoning.

3. The absence of important references. The self-distillation in small models is already studied in [1,2,3].

**Questions:**

**Do self-distillation really work in small model continual training ?**

Fig. 4 in your paper and Fig. 7 in [3] indicated  the performance decreased, when the n of iteration became large. From a theoretical perspective on synthetic data, the data variance decreases with multiple generations n [4]. To prevent variance reduction, i.e., to enhance data diversity, this paper incorporates data synthesized by LLMs throughout the iterative process. This operation is very important. From data perspective, authors should analyze the data distribution shifting across n.

**Is there any new theoretical insights ?**

The authors observed shifts in generation strategies over iterations n. What causes this phenomenon? Additionally, as more data is generated, the overall dataset size increases.

------

[1] Ho N, Schmid L, Yun S Y. Large language models are reasoning teachers[J]. arXiv preprint arXiv:2212.10071, 2022.

[2] Fu Y, Peng H, Ou L, et al. Specializing smaller language models towards multi-step reasoning[C]//International Conference on Machine Learning. PMLR, 2023: 10421-10430.

[3] Zhu X, Qi B, Zhang K, et al. PaD: Program-aided Distillation Can Teach Small Models Reasoning Better than Chain-of-thought Fine-tuning[J]. arXiv preprint arXiv:2305.13888, 2023.

[4] Mobahi H, Farajtabar M, Bartlett P. Self-distillation amplifies regularization in hilbert space[J]. Advances in Neural Information Processing Systems, 2020, 33: 3351-3361.

[5] Dohmatob E, Feng Y, Yang P, et al. A tale of tails: Model collapse as a change of scaling laws[J]. arXiv preprint arXiv:2402.07043, 2024.

---

> ### Author Response · Authors · 2024-11-15
> **Addressing weakness pointed in our work**
>
> We thank the reviewer for providing detailed and easy to follow feedback on our paper. We specially thank the reviewer for citing relevant literature along with the comments. We address the weaknesses and questions highlighted by the reviewer below:
>
> ## Weaknesses
> ### W1: The mathematical notation is overly verbose.
> While some of the notation may look complex, **it is necessary to formulate the iterative self-guided framework in detail, and Algorithm 1 is required to do so**. Furthermore, we use the mathematical framework to define and show the importance of the data mixing rate "alpha", without which it would have been difficult to discuss the two settings: "All" and "Sparse". Finally, the original knowledge distillation paper by Hinton et al. uses KL divergence to discuss distillation effects, and it was important to draw a comparison between SIKeD and KL divergence.
>
> ### W2: Limited generalization
> >The approach only enhances the GSM8K dataset, but reasoning tests should be conducted on more realistic datasets, such as MATH, Arc-Challenge and so on. And more reasoning tasks also need to be evaluated, such as commonsense reasoning and symbolic reasoning.
>
> We wanted to demonstrate the self-iterative knowledge distillation approach where a smaller model can learn to pick the right strategy to solve a given task following LLM. We used Chain of Thought, subquestion decomposition and Program of Thought as our strategies to pick from. Essentially **we were looking for a task that possesses the qualities of intermediate reasoning, can be decomposed into smaller tasks and can be represented in the form of a program. We found mathematical reasoning as the perfect fit for our use case** and other reasoning tasks were hard to fit into the chosen strategies. This is why we limited ourselves to mathematical reasoning and also added that to the title to not oversell our idea. We are collecting a dataset in a different domain that can work across multiple strategies but that is for future work.
>
> Also, **to compare our work with the previous works of knowledge distillation, we limited ourselves to four datasets that were commonly used in the past work** on Magister et al. (https://arxiv.org/abs/2212.08410), Shridhar et al. (https://arxiv.org/abs/2212.00193) and  Zhu et al (https://arxiv.org/abs/2401.11864) which also form our baselines. Finally, we did initial analysis on the MATH dataset and found PoT and L2M to be a weaker strategy compared to CoT which biased the model to always pick CoT.
>
> ### W3: The absence of important references. The self-distillation in small models is already studied in [1,2,3].
> Self-distillation has been studied in the past but our work demonstrates that direct distillation and self-distillation are on the two extremes. A mixture of the two (controlled by alpha in our work) performs better as shown in Fig. 6. In addition, **our work also focuses on teaching multiple strategies to smaller models so that they can learn to pick the right strategy for a given task, which is missing from all previous work on self-distillation**. We will add the highlighted reference in our paper. Thanks for that.

---

> ### Author Response · Authors · 2024-11-15
> **Addressing questions raised by the reviewers**
>
> ## Questions
>
> ### Do self-distillation really work in small model continual training ?
> > Fig. 4 in your paper and Fig. 7 in [3] indicated the performance decreased, when the n of iteration became large. From a theoretical perspective on synthetic data, the data variance decreases with multiple generations n [4]. To prevent variance reduction, i.e., to enhance data diversity, this paper incorporates data synthesized by LLMs throughout the iterative process. This operation is very important. From data perspective, authors should analyze the data distribution shifting across n.
>
> From a theoretical perspective on synthetic data, the data variance decreases with multiple generations as discussed in  [4]. However **[4] uses a model that self-distills itself using a fixed strategy**. On the contrary, **SIKeD uses multiple strategies which allows the model to switch strategies over iterations for a given query, potentially increasing the data variance**. Moreover, we also mix data from LLM in 2 variations: a) All b) Sparse which also changes over iterations. We analyzed the change in strategy distribution in Fig. 2 which shows that **diversity may or may not increase but mixing data reduces overall KL divergence between the training data distribution and the distribution generated by the smaller model leading to better overall performance**.
>
> ### Is there any new theoretical insights ?
> >The authors observed shifts in generation strategies over iterations n. What causes this phenomenon? Additionally, as more data is generated, the overall dataset size increases.
>
> - We observed that smaller models can learn to pick the suitable  strategy for a given task over iterations but this requires careful mixing of data between the LLM and self generated data controlled by the parameter ‘alpha’. **Simply using all the strategy data combined from the LLM wont work** as shown in the Results table (Combined baseline in Table 1).
> - In terms of overall data size, SIKeD increases the data size for ‘All’ variation, but not for ‘Sparse’. The dataset used for training ‘Sparse’ models utilizes correctly generated data by the smaller model, and for questions where the smaller model is unable to generate correct answers, corresponding data from LLM is added. **Over iterations, as the smaller model improves, additional training data from LLM decreases, essentially keeping the training dataset approximately constant. However, we observe that ‘Sparse’ performs as well as the ‘All’ variation**, if not better.
>
> We again thank the reviewer for all the important points raised. Please let us know if all the concerns are addressed. We are happy to discuss more.
> Thank you.

---

> ### Author Response · Authors · 2024-11-18
> **Follow-Up on Rebuttals**
>
> We want to sincerely thank the reviewers for their time and effort in evaluating our paper. We would appreciate it if you could kindly confirm that the rebuttal was received and let us know if any additional steps or clarifications are required from our side. Your feedback is highly important to us, and we remain available to address any further concerns or questions.
>
> Please let us know. Thanks.

---

> ### Comment · Reviewer_YiHw · 2024-11-26
>
> Thank you very much for your clarification. I summarize my concerns as follows:
>
> - **The main contribution overlaps with previous works.** [6] demonstrated that keeping the source real data in iteratively generated data will avoid model collapse. They illustrate this conclusion in both language, vision, and molecular data. They also explain why this happens theoretically. If you keep the source data in your iteration, there is an upper bound for test error. In other words, these conclusions in [6] overlap a lot with your main contribution. Put above in your paper is iterative data mixing. I acknowledge your implementation is important. However, based on previous works, you didn’t provide new conclusions for self-distillation or synthetic data research areas.
> - **Validation is limited compared to previous works.** I know math is suitable, and everyone knows it. However, compared with the classic paper in this line, Chain-of-thought [7] conducted experiments on math, commonsense reasoning, and symbolic reasoning, a total 11 datasets. Your paper only trains on 1 dataset, test on the other four. More recently, [3] also conducted experiments on 7 datasets. I think the validation in this paper is unacceptable, it lacks persuasiveness.
> - **Needing more concise and direct presentation.** I acknowledge that rigorous formalization is important, but what you've done is actually quite straightforward and direct—why not just state it plainly? As I summarized earlier, and I'll reiterate here, ”during the training process using self-distilled data, it is necessary to incorporate the original real data. “ However, as mentioned above, these things have been well-stuied both emperically and theoretically in [6].
>
> **In conclusion, based on overlapping contributions, limited validation, and no theoretical insights, I will keep my rating “3, reject “.** I think this paper needs much improvement to link with the recent literature. Hope my opinions can help you. If I am wrong, please correct me.
>
> [6] Gerstgrasser M, Schaeffer R, Dey A, et al. Is model collapse inevitable? breaking the curse of recursion by accumulating real and synthetic data[J]. arXiv preprint arXiv:2404.01413, 2024.
>
> [7] Wei J, Wang X, Schuurmans D, et al. Chain-of-thought prompting elicits reasoning in large language models[J]. Advances in neural information processing systems, 2022, 35: 24824-24837.

---

> > ### Author Response · Authors · 2024-11-26
> > **Clarification of the concerns raised by the Reviewer**
> >
> > We appreciate the reviewer for initiating this discussion. However, it seems **there has been some misunderstanding regarding our work**, which we would like to address and clarify as follows:
> >
> > ## The main contribution overlaps with the previous work [6]
> >
> > **We respectfully disagree with this comparison, as it equates two fundamentally different approaches**. Below, we detail the key distinctions that clearly differentiate our work from [6]:
> >
> > Firstly, we acknowledge that [6] suggests that combining model-generated data with previously available data can improve model training. This idea has also been explored in other works, such as ReST (https://arxiv.org/abs/2308.08998), which we have already cited in our work. In fact, [6] shares more similarities with ReST than with our work. Similar to ReST, [6] generates new data using the model, combines it with historical data, and retrains the model (referred to as "data accumulation" in [6] and "off-policy training" in ReST). We will update our work to explicitly acknowledge [6] in this context and appreciate the reviewer for pointing it out.
> >
> > Now, we address the core differences between [6] and our work:
> >
> > 1. **Data Combination Strategy**: [6] does not propose a specific method to combine the two datasets, leaving this as an open research question. In contrast, we introduce a data combination parameter, “alpha,” which is automatically determined based on the model's iterative training process.
> > 2. **Training Approach**: [6], like ReST, relies on pre-training or training an initial checkpoint, a computationally expensive process as noted in both papers. Our approach employs on-policy iterative continual training, which is both cost-effective and more efficient.
> > 3. **Data Requirements**: In [6], the data accumulation process results in a growing dataset as old and new data are combined. SIKeD, however, does not require an increase in training data size. Specifically, in the sparse case of our training, only an amount of data equivalent to the original training dataset is added per iteration.
> > 4. **Applicability Across Models**: [6] demonstrates theoretical results only on linear models, with a suggestion that it could be extended to ridge regression and kernel methods. In contrast, our method has been validated across various model sizes and architectures.
> > 5. **Scheduling and Automation**: The authors of [6] themselves note that exploring different schedules for data addition remains future work. We have implemented one such schedule, leveraging the automatically determined alpha parameter, and have demonstrated its effectiveness in Table 1 of our results.
> > 6. **Task Objectives**: Our work tackles a broader and more nuanced goal: enabling smaller models not only to learn better from the data but also to discover appropriate strategies for solving tasks. While an on-policy adaptation of [6] might have limited itself to one fixed strategy, such as Chain-of-Thought (CoT), we aim to show that models can adaptively learn alternate strategies when certain approaches are challenging to learn. Importantly, as shown in our work, naively mixing strategies does not yield effective results. If we were to apply [6]'s approach by mixing strategies indiscriminately, it would fail, as demonstrated.
> >
> > We hope these distinctions clarify the substantial differences between our work and [6]. It is therefore not accurate to claim that our contributions overlap significantly with [6].
> >
> >
> > ## Validation is limited compared to previous works
> >
> > **Our task is not merely about applying reasoning strategies across all possible datasets**. Instead, our focus is on tasks where multiple reasoning strategies (such as CoT, PoT, and L2M) perform equally well. Upon reviewing the CoT, L2M, and PoT papers, as well as the PaD paper, we found that multi-step arithmetic reasoning datasets were the only common datasets investigated across all these works. This limitation is intrinsic to the available benchmarks and is the primary reason why we restricted our experiments to four mathematical reasoning datasets.
> >
> >   - The title of our work, "Self-guided Iterative Knowledge Distillation for Mathematical Reasoning," explicitly specifies the scope of our study. **We do not overclaim generalization to broader reasoning tasks and believe it is reasonable to limit our exploration to four mathematical reasoning datasets**.
> >   - *The L2M paper, published in ICLR, presented results on four datasets*. Similarly, the *PaD paper explored results across three models*. In comparison, **we present results on five different models**. It is therefore not valid to argue that we have three fewer datasets than another work, especially when **we have demonstrated broader model coverage**. As emphasized in our paper, the validation scope (in terms of datasets or models) should align with the specific claims made, which in our case pertain to mathematical reasoning.
> >
> > I hope these points addresses the concerns raised.

---

> > > ### Author Response · Authors · 2024-11-26
> > > **Clarification of the concerns raised by the Reviewer II**
> > >
> > > ## Needing more concise and direct presentation
> > >
> > > We appreciate the reviewer’s suggestion for more concise and direct communication. However, we would like to clarify that our approach goes far beyond merely “mixing data.”
> > >
> > >   - **Our work is not just about combining datasets**. It **leverages self-generated data from the trained model, which is strategically mixed with the original LLM data using an automatically determined parameter**, “alpha.” This parameter is iteratively refined to distill reasoning capabilities into a smaller model effectively.
> > >   - Unlike prior works, **our approach enables smaller models to break free from the limitation of relying on a single reasoning strategy for all tasks** upon distillation. By iteratively training the model using both self-generated and LLM-guided data, we allow the smaller models to learn and adapt to new strategies while consolidating their previous learning.
> > >   - **Unlike computationally expensive methods like ReST or [6], our approach employs on-policy iterative training**. This ensures **computational efficiency** while still improving reasoning performance.
> > >   - We **minimize the KL divergence between the distributions of the LLM and the smaller model**. This enables the **smaller model to adopt reasoning strategies from the LLM effectively, ensuring that new strategies are learned without forgetting previous ones.**
> > >
> > > Calling our paper “just data mixing” is not right in our opinion and we hope that the reviewer will agree to this based on the points that we mentioned above.
> > >
> > > We hope this clarification helps the reviewer appreciate the nuances of our contributions. **We kindly request that the reviewer re-evaluate the score in light of our response and the points presented above**. Thank you again.

---

> > > > ### Comment · Reviewer_YiHw · 2024-12-03
> > > >
> > > > Thank you for your detailed response! I really appreciate the detailed explanation. From your explanation, I have learned a lot.
> > > >
> > > > However, the authors' claim is not convincing.
> > > > My concerns are:
> > > > - limited validation; works in this line are provide more comprehensive results more than 1 dataset training.
> > > > - rigorous mathematical process, but no new theoretical insights (proof or new equation )
> > > > - self-distillation and KL are well-studied existing methods, and this paper contributes a mixing hyperparameter Alpha; but the relationship of real and synthetic data has been well-studied.
> > > >
> > > > **In conclusion, all my concerns stay as it is from the beginning of the rebuttal. So i will keep my score as it is.**

---

### Official Review · Reviewer_BCvZ · 2024-11-04

**Soundness:** 3
**Presentation:** 3
**Contribution:** 3
**Rating:** 6
**Confidence:** 4

**Summary:**

It is hard for a small LLM to learn multiple correct trajectories towards the same ground truth answer, which might affect its reasoning capability, especially the generalization to OOD reasoning tasks. The paper has tested distilling multi-trajectory training data  for post-training smaller LLM from large LLMs and observed unsatisfactory results. To alleviate this problem, the paper proposes a mix of self-generated multi-trajectory training data together with the distilled data for SFT. Surprisingly it achieves better reasoning performance compared to existing methods, especially on OOD tasks.

**Strengths:**

1. intuitive and effective method

2. thorough experimental analysis

**Weaknesses:**

The only concern I  have on the experiment part is that only testing the small model’s preference on COT, POT and L2M is a bit constrained. I’m curious to see that among the three methods,  1)  training with the proposed method, 2) pure distilling  and 3) pure self-generating, which method can make the model generate the most diverse trajectories and whether the diversity is aligned with model’s performance on OOD tasks. Because in each strategy, for example in COT, a model can also generate multiple cot trajectories that lead to the correct answer. I’m curious to see which of the methods can improve the general diversity of the model’s output trajectories the most and whether this diversity is aligned with model’s OOD performance.

**Questions:**

Please see weakness

---

> ### Author Response · Authors · 2024-11-15
> **Addressing reviewers concern**
>
> We thank the reviewer for the positive feedback on our experimental analysis and framework. We discuss the one concern highlighted by the reviewer below:
>
> ## Concern: I’m curious to see that among the three methods, 1) training with the proposed method, 2) pure distilling and 3) pure self-generating, which method can make the model generate the most diverse trajectories and whether the diversity is aligned with model’s performance on OOD tasks
>
> **We refer to Figure 6 which shows that the best performing model is neither a pure distillation (alpha=1) nor pure self-generation (alpha=0), but somewhere in the middle**. SIKeD improves the performance of smaller models by tuning them to select the most accurate reasoning strategy often, but in addition it also enables the smaller model to switch strategies over iterations. Figure 5 and the qualitative analysis (Figure 7) shows this.
>
> From a diversity perspective, pure distillation incorporating all the reasoning strategies (combined in our baseline) has the highest diversity, as a larger model is able to generate more diverse reasoning chains. The diversity of purely self-generating distillation is the lowest, as a smaller model is always biased towards one strategy (Please check Figure 1 for reference).
>
> Hence,  **Diversity(Pure Distillation) > Diversity(SIKeD) > Diversity(Pure Self-Generating)**
>
> However, **Performance(SIKeD) > Performance(Pure Distillation) and Performance(SIKeD) > Performance(Purely Self-generating)**. This is mostly because only using LLM data has a distributional gap with the smaller model while purely using smaller model data has limited correct samples to improve. A mix of both worlds yields the best results as presented in our work.
>
> We are happy to discuss more. Please let us know if any more concerns. Thank you.

---

> ### Author Response · Authors · 2024-11-18
> **Follow-Up on Rebuttals**
>
> We want to sincerely thank the reviewers for their time and effort in evaluating our paper. We would appreciate it if you could kindly confirm that the rebuttal was received and let us know if any additional steps or clarifications are required from our side. Your feedback is highly important to us, and we remain available to address any further concerns or questions.
>
> Please let us know. Thanks.

---

### Official Review · Reviewer_EkQs · 2024-11-04

**Soundness:** 3
**Presentation:** 2
**Contribution:** 2
**Rating:** 6
**Confidence:** 3

**Summary:**

This paper proposes SIKeD, a novel knowledge distillation approach for transferring multistep reasoning skills from large language models (LLMs) to smaller models, particularly for mathematical reasoning tasks. Traditional distillation methods struggle with strategy selection, often resulting in smaller models biased towards a single strategy. SIKeD addresses this by allowing the smaller model to iteratively learn and apply various strategies, combining LLM-generated data with its self-generated outputs to refine strategy selection. The method demonstrates improvements over single-strategy and combined-strategy distillations, achieving superior results across in-distribution and out-of-distribution mathematical reasoning datasets.

**Strengths:**

- The idea is novel. SIKeD leverages the idea of constructivist learning theory and uses an iterative self-guided approach for multi-strategy distillation, compared to previous single-step distillation.
- SIKeD shows generalization across in- and out-of-distribution mathematical reasoning datasets, demonstrating its effectiveness in diverse contexts.
- SIKeD is evaluated using various small model types, showing consistent improvements over the baselines across different model types.
- The paper is well-written and easy to follow.

**Weaknesses:**

- W1: The proposed method is only evaluated on mathematical reasoning tasks. It’s unclear how well SIKeD would generalize to other domains that require more nuanced strategy selection.
- W2: The paper lacks comparison with knowledge distillation methods.
- W3: The code is not available for reproduction.

**Questions:**

- Q1: Have the authors considered applying SIKeD to tasks outside of mathematical reasoning to test the generalizability of the strategy selection mechanism?
- Q2: Does SIKeD require the ground-truth answers of the training data? If not, how does it handle the tasks where the ground-truth answers are not available?
- Q3: Is there any knowledge distillation baseline that could be used to compare the performance of SIKeD on mathematical reasoning tasks? Current experiments only compare SIKeD against CoT, L2M, PoT and Combined.
- Q4: Can the authors explain why does the improvement on Qwen 1.5B model is less significant compared to the other base models? What are the potential reasons for this discrepancy?
- Q5: The small models are tuned with LoRA, what if the parameters of the small models are fully tuned? Would the performance of SIKeD be further improved?
- Q6: "The iterative training is stopped when accuracy shows only marginal improvements or declines." What specific criteria is used to determine the optimal number of iterations?

---

> ### Author Response · Authors · 2024-11-15
> **Addressing weakness pointed in our work**
>
> We thank the reviewer for providing positive feedback for our experimental process, and also for a very clear, detailed and easy to follow list of questionnaires. We address the weaknesses and questions raised by the reviewer below:
>
> ## Weakness1: The proposed method is only evaluated on mathematical reasoning tasks. It’s unclear how well SIKeD would generalize to other domains that require more nuanced strategy selection.
>
> We have addressed this concern in details in the general comment section. In short, we wanted to demonstrate the self-iterative knowledge distillation approach where a smaller model can learn to pick the right strategy to solve a given task following LLM. **We used Chain of Thought, subquestion decomposition and Program of Thought as our strategies to pick from based on the past works**. Essentially we were looking for a task that possesses the qualities of intermediate reasoning, can be decomposed into smaller tasks and can be represented in the form of a program. **We found mathematical reasoning to be the perfect fit for our use case**. The other reasoning tasks were difficult to fit into the chosen strategies. Therefore, we limited ourselves to mathematical reasoning and also added that to the title to not oversell our idea. We are collecting a dataset in a different domain that can work across multiple strategies but that is for future work.
>
> ## Weakness2: The paper lacks comparison with knowledge distillation methods.
> **We compared SIKeD with past knowledge distillation works like CoT based distillation** (CoT in Table 1) which was proposed by Magister et al. (https://arxiv.org/abs/2212.08410), **L2M based distillation** (L2M in Table 1) which was proposed by Shridhar et al. (https://arxiv.org/abs/2212.00193) and **combined** is inspired from Zhu et al (https://arxiv.org/abs/2401.11864). All these comparisons are presented in Table 1 and are considered as baselines against which we compare our proposed methodology SIKeD. We will clarify this in the paper that **our baselines come from previous distillation works**.
>
> ##  The code is not available for reproduction.
> We are currently in the process of cleaning our code and we’ll add a Github link soon.

---

> ### Author Response · Authors · 2024-11-15
> **Addressing reviewers questions.**
>
> Below is a discussion of all the issues the reviewers wanted to discuss.
>
> ## Q1: Have the authors considered applying SIKeD to tasks outside of mathematical reasoning to test the generalizability of the strategy selection mechanism?
> Please refer to the first point under weakness above.
>
> ## Q2: Does SIKeD require the ground-truth answers of the training data? If not, how does it handle the tasks where the ground-truth answers are not available?
> Since the knowledge distillation works with the LLM generating the training data for smaller models, we need to verify that the data generated by the LLM is correct.. If the ground truth data is unavailable, we can either use the LLM to verify the generations of the smaller model or do a self-consistency over the generated samples (since we have K=10) and use the most consistent answer. However, we did not explore these in our work due to the availability of the ground truth data.
>
> ## Q3: Is there any knowledge distillation baseline that could be used to compare the performance of SIKeD on mathematical reasoning tasks? Current experiments only compare SIKeD against CoT, L2M, PoT and Combined
> **Comparison with the baselines of CoT, PoT, L2M and combined are based on the past knowledge distillation works**. CoT based distillation (CoT in Table1) was proposed by Magister et al. (https://arxiv.org/abs/2212.08410), L2M based distillation (L2M in Table 1) was proposed by Shridhar et al. (https://arxiv.org/abs/2212.00193) and the combined one is inspired from Zhu et al (https://arxiv.org/abs/2401.11864). All these comparisons are presented in Table 1 and are considered as baselines against which we compare our proposed methodology SIKeD. We will clarify this in the paper that our baselines come from previous distillation works.
>
> ## Q4: Can the authors explain why does the improvement on Qwen 1.5B model is less significant compared to the other base models? What are the potential reasons for this discrepancy?
> A recent work “Careful Examination of LLM performance on GSM” (https://arxiv.org/abs/2405.00332) has suggested that **some models are over optimized for GSM8K style tasks** and we suspect that could be one of the reasons for minor improvements.
>
> ## Q5: The small models are tuned with LoRA, what if the parameters of the small models are fully tuned? Would the performance of SIKeD be further improved?
> **Experimenting with fully fine-tuning small models showed that the performance worsened, possibly due to over-fitting**. We again refer to “Careful Examination of LLM performance on GSM”” *(https://arxiv.org/abs/2405.00332) which suggests that some models are over-optimized on GSM8K style tasks*
>
> ## Q6: "The iterative training is stopped when accuracy shows only marginal improvements or declines." What specific criteria is used to determine the optimal number of iterations?
> We trained the Gemma 2B model further till 5th iteration and the overall performance went down. We noticed that **training for one or two iterations after the accuracy starts to plateau or goes down is a good stopping criteria**. For our experiments, training up to 3 iterations seems to be a good stopping point.
>
> Please let us know if there are any more concerns. We are happy to discuss more.

---

> ### Author Response · Authors · 2024-11-18
> **Follow-Up on Rebuttals**
>
> We want to sincerely thank the reviewers for their time and effort in evaluating our paper. We would appreciate it if you could kindly confirm that the rebuttal was received and let us know if any additional steps or clarifications are required from our side. Your feedback is highly important to us, and we remain available to address any further concerns or questions.
>
> Please let us know.
> Thanks.

---

### Official Review · Reviewer_JLQW · 2024-11-04

**Soundness:** 3
**Presentation:** 3
**Contribution:** 3
**Rating:** 6
**Confidence:** 3

**Summary:**

The paper presents SIKeD, a knowledge distillation approach to enhance smaller models with reasoning skills from Large Language Models (LLMs). SIKeD employs an iterative process that enables smaller models to learn multiple strategies and select the most appropriate for a given task, addressing the issue of strategy bias found in conventional distillation methods. The paper reports that SIKeD outperforms traditional techniques on mathematical reasoning tasks across various smaller model sizes.

**Strengths:**

1. The paper presents a unique approach to knowledge distillation by introducing the concept of self-guided iterative training. This method allows smaller models to dynamically adjust their strategy preferences, which is a creative solution to the challenge of strategy distribution mismatch in traditional distillation.

2. The experiments are well-designed and conducted across various mathematical reasoning datasets, providing a thorough evaluation of SIKeD's effectiveness. The improvements in performance metrics are substantial and clearly demonstrated.

3. The paper is well-organized, with a clear problem statement and a logical flow of ideas. The methodology is explained in detail, making it easy for readers to understand the proposed approach and its implications.

**Weaknesses:**

1. The paper's methodology, SIKeD, is contingent upon the quality of the initial LLM data. There is a need for further exploration on how fluctuations in LLM data quality might influence the distillation process and the performance of the resulting smaller models.

2. The study primarily focuses on mathematical reasoning tasks, with less clarity on the transferability of SIKeD to other reasoning domains such as commonsense or symbolic reasoning. Additional investigation into the broader applicability of SIKeD could be valuable.

**Questions:**

1. Can the authors comment on the potential of SIKeD to be effective in domains outside of mathematical reasoning? Have there been any preliminary experiments or considerations in this direction?

2. Could the authors elaborate on the computational efficiency of SIKeD, especially in terms of the number of iterations required for convergence and the resources needed for each iteration?

3. The paper discusses various smaller models, but does not discuss how the size of the smaller model affects the distillation process and the final performance. Are there any insights on how SIKeD scales with different model sizes?

---

> ### Author Response · Authors · 2024-11-15
> **Addressing Reviewers concern I**
>
> We thank the reviewer for recognizing and valuing our work, as well as for their positive feedback on our paper writing, presented methodology and experimentation choice.
>
> We address the weaknesses highlighted by the reviewers, along with the questions raised, in detail below:
>
> ## Concern1: SIKeD is contingent upon the quality of the initial LLM data
>
> This is an important point raised and *knowledge distillation as a concept depends on the quality of the teacher model data*. However,  **SIKeD is not limited by the initial LLM data** as SIKeD is a self-guided iterative learning framework. This means that with a weaker LLM, the quality of distillation data will be relatively poor compared to a stronger LLM, leading to a weaker distilled smaller model as baseline. But **our approach improves the baseline performance irrespective of the teacher model used.**
>
> We replace Llama3 70B as LLM with Llama3 8B and report our results for Gemma 2B and 7B models for GSM8K below:
>
> **Gemma-2B**
>
> | Method   | Accuracy with LLaMA3-8B as LLM| Accuracy with LLaMA3-70B as LLM |
> |----------|---------------------------------|----------------------------------|
> | CoT      | 40.79                           | 36.54                           |
> | PoT      | 41.70                           | 44.05                       |
> | L2M      | 37                              |  36.92                         |
> | Combined | 42.08                           | 44.05                           |
> | SIKeD    | **44.35**                       |    **47.23**                   |
>
>
> **Gemma-7B**
>
> | Method   | Accuracy with LLaMA3-8B as LLM | Accuracy with LLaMA3-70B as LLM |
> |----------|---------------------------------|----------------------------------|
> | CoT      | 70.36                           | 67.40                           |
> | PoT      | 67.55                           | 71.34                           |
> | L2M      | 68.99                           | 69.29                           |
> | Combined | 70.66                           | 70.74                           |
> | SIKeD    | **71.04**                       | **73.84**                       |
>
> Overall **SIKeD improved the performance over baselines even when a weaker LLM was used**. We observed similar results with the Qwen models.
>
> ## Concern2: Study primarily focused on mathematical reasoning tasks
>
> We have addressed this concern in details in the general comment section. In short, we used Chain of Thought, subquestion decomposition and Program of Thought as our strategies to pick from based on the past works. Essentially **we were looking for a task that possesses the qualities of intermediate reasoning, can be decomposed into smaller tasks and can be represented in the form of a program. We found mathematical reasoning to be the perfect fit for our use case**. The other reasoning tasks were difficult to fit into the chosen strategies. Therefore, we limited ourselves to mathematical reasoning and also added that to the title to not oversell our idea. We are collecting a dataset in a different domain that can work across multiple strategies but that is for future work.

---

> ### Author Response · Authors · 2024-11-15
> **Addressing Reviewers concern II**
>
> ## Concern3:  Computational efficiency of SIKeD and number of iterations required for convergence
>
> We performed a full Self-guided iterative training using SIKeD and with the “Sparse” version of our approach, we only take the problems from the LLM that the smaller model could not solve. For all queries with incorrect solutions generated by the smaller model, we take the query output pairs from the LLM.
>
> For example, Gemma 7B has a baseline accuracy of ~70%. Assuming the same accuracy on the training dataset (which is true), only 30% of the data from LLM is required for the next iteration. Since all baseline models and each iteration of SIKeD are trained for 3 epochs, training SIKeD with 30% data for 3 epochs (1 iteration of SIKeD) corresponds approximately to 1 epoch of baseline models. Thus, 3 iterations of SIKeD correspond to 3 epochs of baseline models, which is equivalent to 1 additional training round of baseline models. **In other words, the training cost of SIKeD is 2X of the baseline model with no additional test time cost**. Note that we trained the baseline model for 3 more epochs and it led to a worse performance possibly due to overfitting.
>
> **Convergence** - We trained the Gemma 2B model further till 5th iteration and the overall performance went down. We noticed that **training for one or two iterations after the accuracy starts to plateau or goes down is a good stopping criteria. For our experiments, training up to 3 iterations seems to be a good stopping point.**
>
> **Resources Needed** - All the models have been fine-tuned using LoRA using a single RTX GPU with 24 GB RAM.
> In terms of the compute time, here are the training time per model:
>
> **Per iteration**
> - Gemma 2B - 1 hour
> - Qwen 1.5B - 1 hour
> - Qwen 0.5B - 30 mins
> - Gemma 7B - 2 hours
> - SmolLm - 1 hour
>
> **Each training iteration was done under 1 hour (except for Gemma 7B that takes 2 hours per iteration)**. For inference with VLLM, all models except Gemma 7B were run on a single RTX GPU with 24 GB RAM. For inference with Gemma 7B, a single A100 GPU was needed. The inference was completed within 2 hours.
>
>
> ## Concern4:  Scaling of SIKeD with model size
>
> Since all the models used in the paper were initially trained on different datasets (which in most cases are not public), it is hard to judge the scalability of SIKeD with model sizes. Also models of different sizes are often trained on different overall tokens which makes the comparison hard for us to do. Nevertheless, **we found that SIKeD works well across models of all sizes with no significant impact due to scaling the model sizes.**
>
> Please let us know if we addressed your concerns and we are happy to discuss more. Thank you.

---

> ### Author Response · Authors · 2024-11-18
> **Follow-Up on Rebuttals**
>
> We want to sincerely thank the reviewers for their time and effort in evaluating our paper. We would appreciate it if you could kindly confirm that the rebuttal was received and let us know if any additional steps or clarifications are required from our side. Your feedback is highly important to us, and we remain available to address any further concerns or questions.
>
> Please let us know.
> Thanks.

---

### Author Response · Authors · 2024-11-15
**Thanking reviewers and addressing common concerns**

We thank all the reviewers for their feedback on our work.  We thank the reviewers for positive feedback on our experimental analysis (reviewers JLQW, BCvZ, YiHw), novelty and performance of our approach (reviewers JLQW, EkQs) as well as the overall structure of the paper (reviewers JLQW, EkQs, BCvZ, ytP5).

## Below is a summary of our approach:

In our work, **we propose a Self-guided Iterative Knowledge Distillation (SIKeD) approach** for mathematical reasoning tasks. **While Large Language Models can solve a mathematical reasoning task through various strategies, smaller models are often biased on a single strategy** (Figure 1). Simply combining all strategies in a knowledge distillation framework does not work well as the smaller model tends to learn a single strategy well (‘Combined’ baseline in Table 1).

**To solve this problem, SIKeD uses an iterative approach where the self-generated data of SLM is mixed with LLM data in each iteration for training**. The proportion of mixing LLM and SLM data is controlled automatically by the mixing rate ‘alpha’. We compare SIKeD against standard distillation approaches using single as well as multiple reasoning strategies. Our results show that **SIKeD increases the model performance by up to 5 points, with consistent gains across 5 models and 4 mathematical reasoning datasets**.

## We address some of the concerns common across the reviews below -

### Concern1: The study primarily focuses on mathematical reasoning tasks
>(Reviewer JLQW’s Weakness 1 and Question 1, Reviewer EkQs’s Weakness W1 and Question Q1, Reviewer YiHw’s Weakness 2, Reviewer ytP5’s weakness 2)

- We wanted to demonstrate the self-iterative knowledge distillation approach where a smaller model can learn to pick the right strategy to solve a given task following LLM. While it is challenging to find universally applicable datasets for reasoning strategies like CoT, PoT, and L2M, **we have ensured that our baselines align with well-established prior work**. This provides a strong foundation for evaluating SIKeD and underscores its performance improvements over state-of-the-art approaches.
Essentially **we were looking for a task that possesses the qualities of intermediate reasoning, can be decomposed into smaller tasks and can be represented in the form of a program. We found mathematical reasoning to be the perfect fit for our use case**. The other reasoning tasks were difficult to fit into the chosen strategies. Therefore, we limited ourselves to mathematical reasoning and also added that to the title to not oversell our idea. We are collecting a dataset in a different domain that can work across multiple strategies but that is for future work.

- While our current experiments are scoped to mathematical reasoning, we view this as a starting point. Due to the lack of a dataset that can be solved fairly well with strategies such as CoT, L2M and PoT that we explored in the paper, we are limited to mathematical reasoning.  We are actively working on extending SIKeD to other reasoning domains and collecting a new dataset to validate its broader applicability in future work.

### Concern2: Comparison with other knowledge distillation work

To compare our previous works of knowledge distillation, **we use four mathematical datasets that were commonly used in the past work** on Magister et al. (https://arxiv.org/abs/2212.08410), Shridhar et al. (https://arxiv.org/abs/2212.00193) and  Zhu et al (https://arxiv.org/abs/2401.11864). These past works also form our baselines (CoT is taken from Magister et al., L2M is taken from Shridhar et al. and Combined is inspired from Zhu et al. in Table 1). Finally, we did initial analysis on the MATH dataset and found PoT and L2M to be a weaker strategy compared to CoT which biased the model to always pick CoT and hence was not suitable for our work.

*In summary, SIKeD offers a robust solution to enhance smaller model performance through iterative, self-guided knowledge distillation.*

We discuss individual comments in details below.

---

### Meta-Review · Area_Chair_dQ7v · 2024-12-24

**Metareview:**

**Summary:** This paper proposes SIKeD, a Self-guided Iterative Knowledge Distillation approach aimed at improving reasoning capabilities of smaller models by leveraging both large language models (LLMs) and self-generated outputs. The method iteratively adjusts the proportion of teacher and student-generated data, guided by a mixing parameter. The evaluation, primarily conducted on mathematical reasoning datasets, shows improvement over traditional knowledge distillation baselines in both in-distribution and out-of-distribution tasks.

**Decision:** While SIKeD presents a novel iteration-based distillation framework for improving reasoning in smaller models, it falls short in several areas. The evaluation is narrowly focused on mathematical reasoning tasks, lacking broader validation across diverse reasoning domains (JLQW, YiHw, ytP5). Furthermore, the contributions overlap with existing works on iterative data mixing and self-distillation, offering limited methodological novelty (YiHw). The lack of theoretical innovations also somewhat undermines the significance of the proposed approach (YiHw). Lastly, the method’s dependence on the quality of LLM data introduces practical limitations (JLQW).

**Additional Comments On Reviewer Discussion:**

This is a borderline paper. During the discussion phase, reviewers appreciated the authors’ detailed clarifications and additional experimental results. However, fundamental concerns remained unresolved. The authors’ responses did not adequately address overlaps with existing work, nor did they provide convincing theoretical insights or methodological advances. Concerns about  the underlying mechanism of diverse reasoning strategies persisted. These concerns collectively leads to the decision to reject. During the reviewer-AC discussion, no objections were raised to this decision.

---

### Decision · Program_Chairs · 2025-01-22

Reject